# “One-Pot” CuCl_2_-Mediated Condensation/C–S Bond Coupling Reactions to Synthesize Dibenzothiazepines by Bi-Functional-Reagent N, N′-Dimethylethane-1,2-Diamine

**DOI:** 10.3390/molecules27217392

**Published:** 2022-10-31

**Authors:** Dehe Wang, Qichao Lu, Zhanjun Li, Chen Fang, Ran Liu, Bingchuan Yang, Guodong Shen

**Affiliations:** 1School of Chemistry and Chemical Engineering, Liaocheng University, 1 Hunan Avenue, Liaocheng 252000, China; 2Zhejiang Heze Pharmaceutical Technology, 1 Street, Qiantang New District, Hangzhou 310018, China; 3School of Chemistry and Chemical Engineering, Shandong University, Jinan 250100, China; 4The Department of Chemistry, University of South Florida, 4202 East Fowler Avenue, Tampa, FL 33620, USA; 5School of Chemistry and Chemical Engineering, Jinan University, 106 Jiwei Road, Jinan 250022, China

**Keywords:** “one-pot”, CuCl_2_-catalyzed, C–S bond coupling, bi-functional-reagent, DMEDA, dibenzothiazepines

## Abstract

The efficient “One-pot” CuCl_2_-catalyzed C–S bond coupling reactions were developed for the synthesis of dibenzo[*b*,*f*][1,4]thiazepines and 11-methy-ldibenzo[*b*,*f*][1,4]thiazepines via 2-iodobenzaldehydes/2-iodoacetophenones with 2-aminobenzenethiols/2,2′-disulfanediyldianilines by using bifunctional-reagent N, N′-dimethylethane-1,2-diamine (DMEDA), which worked as ligand and reductant. The reactions were compatible with a range of substrates to give the corresponding products in moderate to excellent yields.

## 1. Introduction

Dibenzothiazepine derivatives are a class of molecules with important biological and pharmaceutical activities [1,2,3,4]. For example, Clotiapine (**A**) has good anti-hallucination, the delusion and the anti-excited restlessness function [5]. Quetiapine fumarate (**B**) is effective for both positive symptoms and negative symptoms of schizophrenia, it can also reduce the emotional symptoms associated with schizophrenia, such as depression, anxiety and cognitive deficits [6,7,8]. The 6-Sulfamoyl-10,11-dihydrodibenzo[*b*,*f*][1,4]thiazepine-8-carboxylic acid (**C**) is a structural analogue of nitroxazepine with high activity, indicating its potential medicinal value (Figure 1) [9].

Due to its wide applications, it is meaningful to find some simple, efficient and practical methods for the synthesis of dibenzothiazepines. In past studies, several typical methods have been developed, including the reactions of 2-halobenzaldehyde and 2-aminobenzenethiols or 2,2′-disulfanediyldianiline [10,11,12,13,14], the intramolecular cyclization reactions [9,15], the intramolecular rearrangement reactions [16], and some other methods (Figure 1) [17]. The previous reports have mainly focused on the classic coupling reactions of 2-halogenated benzaldehyde with 2-aminobenzenethiols or 2,2′-disulfanediyldianiline because it is direct and effective. However, the classic copper-catalyzed coupling reactions for the synthesis of dibenzothiazepine are complicated, often required the copper salt, ligand, base and solvent and the substrate application scopes were limited [18]. For example, in 2009, Reiko Yanada et al. employed a Pd(OAc)_2_-catalyzed one-pot reaction to synthesize dibenzo[*b*,*f*][1,4]thiazepines by microwave-accelerated tandem process of 2-brmobenzaldehyde and 2-aminobenzenethiols [19]. In 2015, Yie-jia Cherng et al. also reported the microwave-assisted strategy to assemble dibenzo[*b*,*f*][1,4]thiazepines [10]. In the same year, Anuj Sharma et al. utilized a standard base-catalyzed condensation of amines with aldehydes to afford dibenzo[*b*,*f*][1,4] thiazepines [11]. Recently, Yun Luo and Jiaxi Xu et al. synthesized these heterocyclic molecules using K_2_CO_3_ and ethane-1,2-diol. [20] Copper-catalyzed coupling reactions is a powerful tool for the construction of C–X (X = S, N, O, etc.) bonds and can be applied in the efficient synthesis of heterocyclic compounds in organic synthesis [21,22,23,24,25,26,27]. Simplifying the copper-catalyzed reaction conditions is also an irreplaceable advantage for the application of the reaction. Our group has been working on the copper-catalyzed coupling reactions to synthesize heterocyclic compounds, in continuation of our works to modify the copper-catalyzed cross-coupling reactions [28,29]. Here, we reported an “One-pot” CuCl_2_-catalyzed C–S bond coupling reactions for the synthesis of dibenzo[*b*,*f*][1,4]thiazepines and 11-methyldibenzo[*b*,*f*][1,4]thiazepines via 2-iodobenzaldehydes/2-iodoacetophenones with 2-aminobenzenethiols/2,2′-disulfanediyldianilines by using bi-functional-reagent DMEDA. Compared to the previous reports of synthesizing dibenzothiazepines in entry 1, our research used DMEDA in small quantities as a bifunctional reagent, which worked as ligand and reductant; the substrate scope is wide (the reaction substrates 2-iodobenzaldehydes/2-iodoacetophenones and 2-aminobenzenethiols/2,2′-disulfanediyldianilines could cross-react with each other) and the reaction conditions were simplified, making the reaction conditions more suitable for large-scale production.

## 2. Results

First, 2-iodobenzaldehyde (**1a**) and 2,2′-Disulfanediyldianiline (**1b**) were selected to optimize the reaction conditions (Table 1). The reaction was conducted with CuCl_2_ (15 mol%), **1a** (0.3 mmol), **1b** (0.15 mmol), Cs_2_CO_3_ (0.6 mmol) and 4 Å molecular sieve (25 mg) in DMEDA (0.50 mL) at 110 °C under N_2_ atmosphere for 24 h, the product **1c** was produced in 73% (Table 1, entry 1). When we decreased the amount of DMEDA, 0.25 mL showed the best results (Table 1, entries 1–3). There was a significant decrease without inorganic base Cs_2_CO_3_ (Table 1, entry 4). When K_3_PO_4_ (0.6 mmol) was used for the reaction, **1c** was improved in 82% yield (Table 1, entries 5–6). Then, the reaction temperature was screened, it was found that higher reaction temperature did not change the reaction yield; 110 °C was the best choice (Table 1, entries 7–8). Finally, other copper salts, such as Cu(OAc)_2_, CuSO_4_·5H_2_O, CuI, were surveyed under the conditions, the yields of **1c** were not increased (Table 1, entries 9–11). Without the 4 Å molecular sieves, the reaction yield was also reduced (Table 1, entry 12). A gram-scale reaction was also conducted, the yield of **1c** was the same as entry 5 (Table 1, entry 13). From the above results we could conclude that the bi-functional reagent DMEDA was necessary for the reaction.

With the optimized reaction conditions in hand, the reaction scope was investigated (Table 2). Our initial studies were focused on the reaction of 2-iodobenzaldehydes **a,** with 2,2′-disulfanediyldianilines **b**, and the products **c** could be isolated in moderate-to-good yields. The 2-Bromobenzaldehyde and 2-chlorobenzaldehyde were used to react with **1b**, and the yields decreased. When 2,2′-disulfanediyldianilines bearing electron-donating and electron-withdrawing groups were used to react with **1a**, the products were obtained in good yields (Table 2, entries 2–4). Substituted 2-iodobenzaldehydes were also used to react with **1b**, and the reactions yields were basically kept in good yields (Table 2, entries 5–9). Some cross-reactions were tested, and moderate-to-excellent yields were obtained. (Table 2, entries 10–12). The above results indicated that the reaction yields of 2-iodobenzaldehydes **a** with 2,2′-disulfanediyldianilines **b** were not influenced significantly by the electronic effect and steric effect. Subsequently, we examined the reaction of 2-iodobenzaldehydes **a** with 2-aminobenzenethiols **b**′; the reactions proceeded smoothly, and the reaction yields were obtained in moderate-to-good yields (Table 2, entries 16–22).

The scope of 2′-iodoacetophenones **d** and 2,2′-disulfanediyldianilines **b**/2-aminobenzenethiol **1b′** was also investigated (Table 3). The 2′-Iodoacetophenone **1d** and 2,2′-disulfanediyldianiline **1b** were used under the optimal conditions, and 75% yield of **1e** was isolated (Table 3, entry 1). We used 2′-iodoacetophenones **1d** and substituted 2,2′-disulfanediyldianilines **2b**–**5b** to do the reaction, and the products **e** were isolated in moderate-to-good yields (Table 3, entries 2–5). Then, 2,2′-disulfanediyldianiline **1b** was replaced by 2-aminobenzenethiols **1b′** and reacted with 2′-iodoacetophenone **1d** to obtain the desired product in 95% yield (Table 3, entry 6). Finally, (2-iodophenyl)(phenyl)methanone **3d** was used, and it could not react with 1b to generate the product **6e** (Table 3, entry 7).

Finally, a possible mechanism was proposed for the reaction based on the experimental results (Figure 2). Firstly, CuCl_2_ coordinates with DMEDA and is reduced to generate the Cu(I) complex (**A**). At the same time, the starting material **1a** could react with **1b** to form the intermediate (**B**) through the intermolecular condensation. The intermediate (**C**) could be generated via the oxidative addition of (**A**) and (**B**). Then, (**C**) was converted to the intermediate (**D**), and DMEDA might act as the reductant. For the substrate **1a** and **1b′**, the intermediate (**D**) is also generated during the reaction process. After the transmetallation/reductive elimination reaction, the product **1c** could be formed. From the possible mechanism, we found that the bifunctional reagent DMEDA worked as ligand and reductant.

## 3. Experimental Section

### 3.1. General

^1^H NMR and ^13^C NMR spectra were recorded 500 MHz (Bruker, Kanton Zug, Switzerland) instrument; CDCl_3_ (δ_H_ = 7.26 ppm, δ_C_ = 77.16 ppm) was used as the internal standard. Chemical shifts were reported in ppm. Multiplicity was recorded: s (singlet), d (doublet), t (triplet), q (quartet), dd (doublet of doublets), dt (doublet of triplets), m (multiplet). The direct used reagents and solvents were pure analytical grade and purchased from commercial sources, if not stated otherwise. The starting substrates were synthesized according to the known literature. Column chromatography was hand packed with silica gel (200–300 mesh). The melting points were uncorrected. High-resolution mass spectra (HRMS) were recorded on a Q-TOF Premier (ESI, Waters, Milford, CT, USA). The silica gel plates (GF254, 0.2 mm thick) were used for TLC testing. 

### 3.2. General Procedure for the Synthesis of Dibenzo[b,f][1,4]thiazepines (**1c**–**12c**) Catalyzed by CuCl_2_ in DMEDA

An oven-dried 25 mL flask equipped with a rubber stopper was charged with a magnetic stir bar, CuCl_2_ (15 mol%, 0.045 mmol), 2-iodobenzaldehyde **a** (0.3 mmol), 2,2′-disulfanediyldianilines **b** (0.15 mmol)/ 2-aminobenzenethiols **b′** (0.3 mmol), K_3_PO_4_ (0.6 mmol), 4 Å molecular sieve (25 mg) and DMEDA (0.5 mL). The reaction mixture was stirred at 110 °C for 24 h. The reaction was monitored by TLC. When benzaldehydes a was consumed, the reaction was stopped and cooled to room temperature, the crude reaction mixture was diluted with 20 mL water, extracted with ethyl acetate (20 mL × 3), combined with organic phase, then washed organic phase with brine (20 mL), dried organic phase with anhydrous Mg_2_SO_4_. The organic phase was concentrated and the residue was purified directly by column chromatography on silica gel using petrol/EtOAc as eluent to give the pure products **c**. 

### 3.3. General Procedure for the Synthesis of 11-Methyldibenzo[b,f][1,4]thiazepines (**1e**–**5e**) Catalyzed by CuCl_2_ in DMEDA

An oven-dried 25 mL flask equipped with a rubber stopper was charged with a magnetic stir bar, CuCl_2_ (15 mol%, 0.045 mmol), 1-(2-iodophenyl)ethan-1-ones **d** (0.3 mmol), 2,2′-disulfanediyldianilines **b** (0.15 mmol)/2-aminobenzenethiols **b′** (0.3 mmol), K_3_PO_4_ (0.6 mmol), 4 Å molecular sieve (25 mg) and DMEDA (0.5 mL). The reaction mixture was stirred at 110 °C for 24 h. The reaction was monitored by TLC. When benzaldehydes d was consumed, the reaction was stopped and cooled to room temperature, the crude reaction mixture was diluted with 20 mL water, extracted with ethyl acetate (20 mL × 3), combined with organic phase, then washed organic phase with brine (20 mL), dried organic phase with anhydrous Mg_2_SO_4_. The organic phase was concentrated and the residue was purified directly by column chromatography on silica gel using petrol/EtOAc as eluent to give the pure products **e**. For the NMR spectrum of compounds see the Appendix A.

### 3.4. Characterization Data

The dibenzo[*b*,*f*][1,4]thiazepine **1c** (Flash column chromatography on silica gel using petrol/EtOAc (6:1, *v*:*v*) as eluent). Yellow solid; mp: 126–128 °C (Lit: m.p. 124 °C) [30]; 52.0 mg; 82% yield (**1a** and **1b** were used); 51.3 mg; 81% yield (**1a** and **1b′** were used); ^1^H NMR (500 MHz, CDCl_3_/TMS): δ 8.90 (s, 1H), 7.44–7.30 (m, 7H), 7.19–7.15 (m, 1H). ^13^C NMR (125 MHz, CDCl_3_/TMS): δ 162.4, 148.7, 139.5, 137.4, 132.9, 131.8, 131.6, 129.5, 129.4, 129.0, 128.4, 127.3, 127.1.

The 7-methyldibenzo[*b*,*f*][1,4]thiazepine **2c** (Flash column chromatography on silica gel using petrol/EtOAc (6:1, *v*:*v*) as eluent). Yellow solid; mp: 115–117 °C; 54.0 mg; 80% yield (**1a** and **2b** were used); 54.0 mg; 80% yield (**1a** and **2b′** were used); ^1^H NMR (500 MHz, CDCl_3_/TMS): δ 8.85 (s, 1H), 7.42 (d, *J* = 7.5 Hz, 1H), 7.39–7.33 (m, 3H), 7.24–7.20 (m, 2H), 7.13 (d, *J* = 8.0 Hz, 1H), 2.30 (s, 3H). ^13^C NMR (125 MHz, CDCl_3_/TMS): δ 161.8, 146.3, 139.3, 137.6, 137.5, 133.2, 131.7, 131.5, 130.2, 129.5, 128.5, 128.3, 127.0, 20.7.

The 7-methoxydibenzo[*b*,*f*][1,4]thiazepine **3c** (Flash column chromatography on silica gel using petrol/EtOAc (5:1, *v*:*v*) as eluent). Yellow solid; mp: 101–102 °C; 62.2 mg; 86% yield (**1a** and **3b** were used); 62.2 mg; 86% yield (**1a** and **3b′** were used); ^1^H NMR (500 MHz, CDCl_3_/TMS): δ 8.80 (s, 1H), 7.43–7.34 (m, 4H), 7.24 (s 1H), 6.95 (d, *J* = 3.0 Hz, 1H), 6.88 (dd, *J*_1_ = 9.0 Hz, *J*_2_ = 3.0 Hz, 1H), 3.79 (s, 3H). ^13^C NMR (125 MHz, CDCl_3_/TMS): δ 160.8, 159.1, 142.4, 138.6, 137.5, 131.7, 131.5, 129.6, 129.4, 128.5, 128.4, 117.0, 115.6, 55.7. HRMS (ESI): *m*/*z* calcd for C_14_H_12_NOS [M + H]^+^: 242.0634, found: 242.0641.

The 7-chlorodibenzo[*b*,*f*][1,4]thiazepine **4c** (Flash column chromatography on silica gel using petrol/EtOAc (6:1, *v*:*v*) as eluent). Yellow solid; mp: 75–76 °C; 65.6 mg; 89% yield (**1a** and **4b** were used); ^1^H NMR (500 MHz, CDCl_3_/TMS): δ 8.87 (s, 1H), 7.44–7.36 (m, 5H), 7.28 (dd, *J*_1_ = 8.5 Hz, *J*_2_ = 2.5 Hz, 1H), 7.23 (d, *J* = 8.5 Hz, 1H). ^13^C NMR (125 MHz, CDCl_3_/TMS): δ 162.7, 147.2, 138.6, 137.2, 132.8, 132.3, 131.90, 131.88, 130.3, 129.6, 129.5, 128.7, 128.0.

The 3-methyldibenzo[*b*,*f*][1,4]thiazepine **5c** (Flash column chromatography on silica gel using petrol/EtOAc (6:1, *v*:*v*) as eluent) Yellow solid; mp: 93–94 °C; 56.7 mg; 84% yield (**2a** and **1b** were used); 52.8 mg; 78% yield (**2a** and **1b′** were used); ^1^H NMR (500 MHz, CDCl_3_/TMS): δ 8.86 (s, 1H), 7.42–7.40 (m, 1H), 7.33–7.28 (m, 3H), 7.20–7.14 (m, 3H), 2.33 (s, 3H). ^13^C NMR (125 MHz, CDCl_3_/TMS): δ 162.5, 148.7, 138.5, 137.2, 136.2, 132.8, 132.4, 131.6, 130.1, 129.29, 129.28, 127.2, 127.0, 21.1.

The 2-methyldibenzo[*b*,*f*][1,4]thiazepine **6c** (Flash column chromatography on silica gel using petrol/EtOAc (6:1, *v*:*v*) as eluent). Yellow solid; mp: 92–94 °C; 49.3 mg; 73% yield (**3a** and **1b** were used); 56.1 mg; 83% yield (**3a** and **1b′** were used); ^1^H NMR (500 MHz, CDCl_3_/TMS): δ 8.86 (s, 1H), 7.42–7.40 (m, 1H), 7.33–7.28 (m, 3H), 7.20–7.14 (m, 3H), 2.34 (s, 3H). ^13^C NMR (125 MHz, CDCl_3_/TMS): δ 162.5, 148.8, 138.5, 137.2, 136.3, 132.8, 132.4, 131.6, 130.1, 129.33, 129.28, 127.2, 127.1, 21.1.

The 3-chlorodibenzo[*b*,*f*][1,4]thiazepine **7c** (Flash column chromatography on silica gel using petrol/EtOAc (6:1, *v*:*v*) as eluent). White solid; mp: 107–109 °C; 63.3 mg; 86% yield (**4a** and **1b** were used); 64.7 mg; 88% yield (**4a** and **1b′** were used); ^1^H NMR (500 MHz, CDCl_3_/TMS): δ 8.84 (s, 1H), 7.45 (d, *J* = 1.0 Hz 1H), 7.41 (dd, *J*_1_ = 7.5 Hz, *J*_2_ = 1.0 Hz, 1H), 7.37–7.29 (m, 4H), 7.19 (td, *J*_1_ = 7.0 Hz, *J*_2_ = 1.5 Hz, 1H). ^13^C NMR (125 MHz, CDCl_3_/TMS): δ 161.2, 148.6, 141.1, 137.8, 135.7, 133.0, 131.6, 130.5, 129.7, 128.6, 128.2, 127.6, 127.2.

The 3-fluorodibenzo[*b*,*f*][1,4]thiazepine **8c** (Flash column chromatography on silica gel using petrol/EtOAc (6:1, *v*:*v*) as eluent). Yellow solid; mp: 54–56 °C; 53.5 mg; 78% yield (**5a** and **1b** were used); 61.1 mg; 89% yield (**5a** and **1b′** were used); ^1^H NMR (500 MHz, CDCl_3_/TMS): δ 8.84 (s, 1H), 7.42–7.30 (m, 4H), 7.22–7.15 (m, 2H), 7.05 (td, *J*_1_ = 8.5 Hz, *J*_2_ = 3.0 Hz, 1H). ^13^C NMR (125 MHz, CDCl_3_/TMS): δ 164.4 (d, *J* = 253.6 Hz), 161.2, 148.6, 141.8 (d, *J* = 8.3 Hz), 133.7 (d, *J* = 3.4 Hz), 133.0, 131.4 (d, *J* = 9.3 Hz), 129.7, 128.2, 127.5, 127.1, 118.8 (d, *J* = 22.3 Hz), 115.6 (d, *J* = 21.8 Hz).

The 2-chlorodibenzo[*b*,*f*][1,4]thiazepine **9c** (Flash column chromatography on silica gel using petrol/EtOAc (6:1, *v*:*v*) as eluent). Yellow solid; mp: 111–112 °C; 61.2 mg; 83% yield (**6a** and **1b** were used); 58.9 mg; 80% yield (**6a** and **1b′** were used); ^1^H NMR (500 MHz, CDCl_3_/TMS): δ 8.82 (s, 1H), 7.41 (dd, *J*_1_ = 8.0 Hz, *J*_2_ = 1.5 Hz, 1H), 7.37–7.29 (m, 5H), 7.18 (td, *J*_1_ = 7.5 Hz, *J*_2_ = 1.5 Hz, 1H). ^13^C NMR (125 MHz, CDCl_3_/TMS): δ 160.7, 148.5, 138.5, 138.0, 134.7, 133.0, 132.9, 131.5, 129.7, 129.3, 128.5, 127.6, 127.1.

The 7-methoxy-3-methyldibenzo[*b*,*f*][1,4]thiazepine **10c** (Flash column chromatography on silica gel using petrol/EtOAc (5:1, *v*:*v*) as eluent). Yellow solid; mp: 128–129 °C; 56.8 mg; 74% yield (**2a** and **3b** were used); ^1^H NMR (500 MHz, CDCl_3_/TMS): δ 8.76 (s, 1H), 7.29 (d, *J* = 8.5 Hz, 1H), 7.23 (d, *J* = 8.5 Hz, 1H), 7.18 (d, *J* = 7.0 Hz, 2H), 6.94 (d, *J* = 3.0 Hz, 1H), 6.87 (dd, *J*_1_ = 8.5 Hz, *J*_2_ = 2.5 Hz, 1H), 3.78 (s, 3H), 2.33 (s, 3H). ^13^C NMR (125 MHz, CDCl_3_/TMS): δ 160.9, 159.0, 142.5, 138.6, 137.3, 135.2, 132.3, 131.6, 130.1, 129.7, 128.4, 116.9, 115.5, 55.7, 21.1. HRMS (ESI): *m*/*z* calcd for C_15_H_14_NOS [M + H]^+^: 256.0791, found: 256.0794. 

The 2-chlorodibenzo[*b*,*f*][1,4]thiazepine **11c** (Flash column chromatography on silica gel using petrol/EtOAc (6:1, *v*:*v*) as eluent). Gray solid; mp: 110–111 °C; 65.4 mg; 84% yield (**6a** and **2b** were used); 60.8 mg; 78% yield (**6a** and **2b′** were used); ^1^H NMR (500 MHz, CDCl3/TMS): δ 8.77 (s, 1H), 7.36–7.32 (m, 3H), 7.24–7.19 (m, 2H), 7.16–7.13 (m, 1H), 2.30 (s, 3H). ^13^C NMR (125 MHz, CDCl_3_/TMS): δ 160.1, 146.1, 138.5, 137.9, 137.8, 134.6, 133.2, 132.9, 131.4, 130.5, 129.3, 128.0, 127.1, 20.7. HRMS (ESI): *m*/*z* calcd for C_14_H_11_ClNS [M + H]^+^: 260.0295, found: 260.0284.

The 2-chloro-7-methoxydibenzo[*b*,*f*][1,4]thiazepine **12c** (Flash column chromatography on silica gel using petrol/EtOAc (5:1, *v*:*v*) as eluent). Yellow solid; mp: 111–112 °C; 62.0 mg; 75% yield (**6a** and **3b** were used); 66.1 mg; 80% yield (**6a** and **3b′** were used); ^1^H NMR (500 MHz, CDCl_3_/TMS): δ 8.71 (s, 1H), 7.34 (s, 3H), 7.25 (d, *J* = 8.7 Hz, 1H), 6.94 (d, *J* = 2.8 Hz, 1H), 6.90 (dd, *J*_1_ = 2.8 Hz, *J*_2_ = 8.8 Hz, 1H), 3.79 (s, 3H). ^13^C NMR (125 MHz, CDCl_3_/TMS): δ 159.3, 159.0, 142.2, 138.6, 137.0, 134.8, 133.0, 131.4, 129.3, 128.9, 128.6, 117.1, 115.8, 55.8. HRMS (ESI): *m*/*z* calcd for C_14_H_11_ClNOS [M + H]^+^: 276.0244, found: 276.0235. 

The 11-methyldibenzo[*b*,*f*][1,4]thiazepine **1e** (Flash column chromatography on silica gel using petrol/EtOAc (6:1, *v*:*v*) as eluent). Yellow solid; mp:75–76 °C (Lit: m.p. 76 °C) [31]; 50.0 mg; 75% yield (**1d** and **1b** were used); 64.2 mg; 95% yield (**1d** and **1b′** were used); ^1^H NMR (500 MHz, CDCl_3_/TMS): δ 7.46–7.45 (m, 1H), 7.42–7.40 (m, 2H), 7.34–7.30 (m, 2H), 7.28–7.24 (m, 1H), 7.18 (dd, *J*_1_ = 1.3 Hz, *J*_2_ = 8.0 Hz, 1H), 7.05 (td, *J*_1_ = 1.3 Hz, *J*_2_ = 7.5 Hz, 1H), 2.66 (s, 3H). ^13^C NMR (125 MHz, CDCl_3_/TMS): δ 169.9, 148.8, 140.0, 139.5, 132.5, 132.0, 130.8, 129.2, 128.9, 128.5, 128.0, 125.6, 125.4, 29.6.

The 7,11-dimethyldibenzo[*b*,*f*][1,4]thiazepine **2e** (Flash column chromatography on silica gel using petrol/EtOAc (6:1, *v*:*v*) as eluent) Yellow solid; mp: 115–117 °C; 53.1 mg; 74% yield (**1d** and **2b** were used); ^1^H NMR (500 MHz, CDCl_3_/TMS): δ 7.46–7.43 (m, 1H), 7.42–7.38 (m, 1H), 7.33–7.29 (m, 2H), 7.23 (s, 1H), 7.09–7.05 (m, 2H), 2.65 (s, 3H), 2.27 (s, 3H). ^13^C NMR (125 MHz, CDCl_3_/TMS): δ 169.3, 146.4, 139.9, 139.5, 135.6, 132.8, 131.9, 130.7, 130.1, 128.40, 128.39, 128.0, 125.3, 29.6, 20.7.

The 7-methoxy-11-methyldibenzo[*b*,*f*][1,4]thiazepine **3e** (Flash column chromatography on silica gel using petrol/EtOAc (5:1, *v*:*v*) as eluent). Yellow solid; mp: 130–132 °C; 52.9 mg; 69% yield (**1d** and **3b** were used); ^1^H NMR (500 MHz, CDCl_3_/TMS): δ 7.46–7.43 (m, 1H), 7.42–7.39 (m, 1H), 7.34–7.30 (m, 2H), 7.12 (d, *J* = 8.9Hz, 1H), 6.96 (d, *J* = 2.9 Hz, 1H), 6.83 (dd, *J*_1_ = 2.8 Hz, *J*_2_ = 8.8 Hz, 1H), 3.76 (s, 3H), 2.64 (s, 3H). ^13^C NMR (125 MHz, CDCl_3_/TMS): δ 168.5, 157.6, 142.5, 139.6, 139.4, 132.0, 130.7, 129.2, 128.5, 128.0, 126.6, 116.6, 115.7, 55.7, 29.5. HRMS (ESI): *m*/*z* calcd for C_15_H_14_NOS [M + H]^+^: 256.0791, found: 256.0794.

The 7-chloro-11-methyldibenzo[*b*,*f*][1,4]thiazepine **4e** (Flash column chromatography on silica gel using petrol/EtOAc (6:1, *v*:*v*) as eluent). Yellow solid; mp: 115–117 °C; 47.5 mg; 61% yield (**1d** and **4b** were used); ^1^H NMR (500 MHz, CDCl_3_/TMS): δ 7.47–7.44 (m, 1H), 7.43–7.39 (m, 2H), 7.36–7.33 (m, 2H), 7.21 (dd, *J*_1_ = 2.3 Hz, *J*_2_ = 8.6 Hz, 1H), 7.10 (d, *J* = 8.5 Hz, 1H), 2.65 (s, 3H). ^13^C NMR (125 MHz, CDCl_3_/TMS): δ 170.4, 147.4, 139.3, 139.2, 132.2, 131.9, 131.0, 130.8, 130.1, 129.3, 128.8, 128.0, 126.4, 29.6. HRMS (ESI): *m*/*z* calcd for C_14_H_11_ClNS [M + H]^+^: 260.0295, found: 260.0275.

The 2,3,9-trimethoxy-11-methyldibenzo[*b*,*f*][1,4]thiazepine **5e** (Flash column chromatography on silica gel using petrol/EtOAc (3:1, *v*:*v*) as eluent). Yellow solid; mp: 52–54 °C; 72.9 mg; 77% yield (**2d** and **5b** were used); ^1^H NMR (500 MHz, CDCl_3_/TMS): δ 7.10 (d, *J* = 8.8 Hz, 1H), 6.95–6.90 (m, 2H), 6.86 (s, 1H), 6.83 (dd, *J*_1_ = 2.8 Hz, *J*_2_ = 8.8 Hz, 1H), 3.87 (d, *J* = 10.5 Hz, 6H), 3.76 (s, 3H), 2.62 (s, 3H). ^13^C NMR (125 MHz, CDCl_3_/TMS): δ 167.9, 157.5, 150.9, 149.3, 142.5, 132.0, 130.9, 129.7, 126.5, 116.4, 115.6, 114.3, 110.7, 56.3, 56.2, 55.7, 29.3. HRMS (ESI): *m*/*z* calcd for C_17_H_18_NO_3_S [M + H]^+^: 316.1002, found: 316.1012.

## 4. Conclusions

In summary, this paper reported a ‘one-pot’ CuCl_2_ catalyzed synthesis of dibenzo[*b*,*f*][1,4]thiazepines and 11-methyldibenzo[*b*,*f*][1,4]thiazepines via the condensation/C–S bond coupling reactions of 2-iodobenzaldehydes/2-iodoacetophenones with 2-aminobenzenethiols/2,2′-disulfanediyldianilines in moderate-to-good yields. The reaction is easy to operate, uses readily available and bi-functional-reagent DMEDA working as ligand and reductant, and exhibits functional group tolerance.

## Data Availability

The data used to support the findings of this study are available from the corresponding author upon request.

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
