# Peer review of "“One-Pot” CuCl2-Mediated Condensation/C–S Bond Coupling Reactions to Synthesize Dibenzothiazepines by Bi-Functional-Reagent N, N′-Dimethylethane-1,2-Diamine"

_molecules, 2022, doi:10.3390/molecules27217392_

Round 1

Reviewer 1 Report (Previous Reviewer 1)

A new submitted version of manuscript molecules-1950531 has been reviewed. The manuscript has greatly been improved in the process of editing. While I am still not completely convinced that the manuscript describes sufficiently significant findings, I can now understand clearly what the authors want to state. Possibly, I am not completely aware of the significance of C-S bond coupling reactions to synthesize Dibenzothiazepines and given the other reviewers positive comments I am now are also able to support acceptance of the manuscript.

Author Response

Thanks for your support on the acceptance of the manuscript, you have made much valuable and constructive suggestions for the revision of the manuscript.

Reviewer 2 Report (Previous Reviewer 3)

The current version of the submission practically does not differ from the previously (several phrases) rejected paper. In my opinion, research possesses scientific interest, but authors have failed in novelty and results presentation. Furthermore, the level of language is too low. 

In point of view research, the suggested mechanism seems to be incorrect. For example, the presence of six-coordinated copper atoms, the suggestion of reducing DMEDa as well as the insertion of Cu into the C-I bond is highly doubtful. 

If the paper was rejected the first time, I can not find any reasons for its publication now.

Author Response

We have carefully revised the manuscript, especially the potential reaction mechanism, the changes were highlighted by using the “Track Changes” function. From the revised mechanism, the copper was two-coordinated atom. And the insertion of activated cuprous salt into the C-H bond is common for the Ullmann type-coupling reactions.

Reviewer 3 Report (New Reviewer)

““One-pot” CuCl2-mediated Condensation/C-S Bond Coupling 2 Reactions to Synthesize Dibenzothiazepines by Bi-functional-3 reagent N, N’-dimethylethane-1,2-diamine”

The author described the CuCl2-mediated condensation for the synthesis dibenzothiazepines. The role of DMEDA and the substrate scope using ketone are interesting. However, the reviewer thinks that further study would be required for the publication in Molecules.

1. The authors should perform the reaction using DMEDA oxide as a control.

2. The reaction with ketone is very interesting. Is it possible to use various ketones such as propiophenone, butyrophenone, and benzophenone ? Further results concerning ketones would be informative for readers.

Author Response

  1. The authors should perform the reaction using DMEDA oxide as a control.

Response: Thanks for your suggestion, we added the DMEDA oxide to repeat the reaction under the optimal reaction condition, the reaction yield was not changed, it indicated that the DMEDA oxide has no inhibitory effect for the reaction.

  1. The reaction with ketone is very interesting. Is it possible to use various ketones such as propiophenone, butyrophenone, and benzophenone? Further results concerning ketones would be informative for readers.

Response: Thanks for your insightful suggestion. Unfortunately, when propiophenone, butyrophenone, benzophenone and (2-iodophenyl)(phenyl)methanone were used to do the reactions, the corresponding dibenzothiazepines could not be detected. Some reaction data were added in the manuscript. We will study the related research to synthesize the dibenzothiazepines by the C-H bond reaction conditions.

Round 2

Reviewer 2 Report (Previous Reviewer 3)

The paper could be published in the present form

Reviewer 3 Report (New Reviewer)

This reviewer has communicated to the editor they now recommend publication in Molecules

This manuscript is a resubmission of an earlier submission. The following is a list of the peer review reports and author responses from that submission.

Round 1

Reviewer 1 Report

The work reported by Yang and Shen group describes the “one pot” CuCl2 catalyzed synthesis of Dibenzothiazepine derivatives. As the authors mention in the introduction there are several reports known for the synthesis of Dibenzothiazepine derivatives. The similar study was reported by Sharma et al. (Ref 11), where they used CuI as catalyst with almost similar reaction condition. The only novelty is using ketone derivatives instead of aldehydes, but the substrate scope is limited. This reviewer will consider it for publication if the authors provide unique elaboration of this work. Therefore, I believe the above study does not bear any outstanding significance that could be considered for publication in Molecules.

Author Response

Response: Thanks for your insightful suggestion. Compared to the previous reoprts of using CuI as catalyst to synthesize dibenzothiazepine derivatives, our research used N, N’-dimethylethane-1,2-diamine (DMEDA) in small quantities as bifunctional-reagent, which worked as ligand and reductant, the reaction range is wide (the reaction substrates 2-iodobenzaldehydes/2-iodoacetophenones and 2-aminobenzenethiols/2,2'-disulfanediyldianilines could cross-react with each other) and the reaction conditions were simplified, is suitable for organic synthesis. We added the gram-scale (5 mmol scale) experiment to repeat the reaction in the manuscript, and the small amount of bifunctional-reagent DMEDA could be recycled in 71% by distillation of the reaction mixtrue.

Reviewer 2 Report

The present article by Dehe Wang et al describes an efficient “one-pot” copper chroride catalyzed synthesis of dibenzothiazepine derivatives, which represent a bioactive scaffold. The reaction conditions were optimized and the scope of the developed reaction was studied. Overall it is a well written paper which could be of interest to the researchers in the field. In revising their manuscript the authors should take under consideration the following:

1. Line 71, “b The yields were determined by 1H NMR analysis.” This should be briefly explained.

2. Line 132, “2'-iodoacetophenones 1d” should change to “2'-iodoacetophenones 1d and 2d”.

3. Line 133, “disulfanediyldianilines” should change to “disulfanediyldianilines 2b-5b”.  

4. Line 134, “(entries 2-6) should change to “(entries 2-5)”.  

5. Line 136, “(entry 8) should change to “(entry 6)”.     

Author Response

  1. Line 71, “b The yields were determined by 1H NMR analysis.” This should be briefly explained.

Response: In Line 71, b The yields were determined by 1H NMR analysis.” was briefly explained as “b The yields were determined by 1H NMR analysis using 1,3,5-Trimethoxybenzene as the internal standard”.

  1. Line 132, “2'-iodoacetophenones 1d” should change to “2'-iodoacetophenones 1d and 2d”.

Response: In Line 132, 2'-Iodoacetophenone 1d reacted with 2,2'-disulfanediyldianiline 1b under the optimal conditions (Table 3, entry 1).

  1. Line 133, “disulfanediyldianilines” should change to “disulfanediyldianilines 2b-5b”.

Response: In Line 133, “disulfanediyldianilines” was change to “disulfanediyldianilines 2b-5b”.

  1. Line 134, “(entries 2-6) should change to “(entries 2-5)”.

Response:  In Line 134, “(entries 2-6)” was change to “(entries 2-5)”.

  1. Line 136, “(entry 8)” should change to “(entry 6)”.

Response: In Line 136, “(entry 8)” was change to “(entry 6)”

Reviewer 3 Report

The paper by Yang and Shen et al. represents the synthesis of dibenzothiazepines by CuCl2-mediated C-S Bond coupling reactions. The work is an increment in the sphere of such a type of reaction. The paper could be suggested for publication in Molecules only after careful reworking:

1.           The order of presentation of tables, diagrams and figures and text is not correct. The text should be first. It isn't easy to read.

2.           What is the difference between entry 1 and this work on the scheme? The scheme demonstrates the reactions. The authors should explain the difference between the data presented, with a description of the catalysts/reaction conditions. Without these, the work is meaningless.

3. Scheme 1 includes a bromine derivative, but the entire work is based on iodine derivatives. In the case of bromine, there is only one example that has a low yield. In this case, the result does not make sense, because the cited literature presents similar fluorine, chlorine, and bromine derivatives with good yields.

4. The decryption of DMEDA looks necessary in the main text of the article, at the first mention.

5. If the DMEDA is a bifunctional reagent, its role must be described. Moreover, this reagent is presented as a key in a catalytic process including the title. The acidic catalysis is more typical of the Schiff reaction. The entire process is not entirely clear and a plausible mechanism is suggested to be given. In addition, why are molecular sieves used? It is necessary to show the results of the reaction without cucl2 in the presence of molecular sieves.

 6. Authors should check references, for example, ref. 15 - journal name should be Chem. Eur. J. In addition, check the writing of the Refs. 30.31.

Author Response

  1. The order of presentation of tables, diagrams and figures and text is not correct. The text should be first. It isn't easy to read.

Response: Thanks for your insightful suggestion. The text was moved in front of the tables, diagrams and figures.

  1. What is the difference between entry 1 and this work on the scheme? The scheme demonstrates the reactions. The authors should explain the difference between the data presented, with a description of the catalysts/reaction conditions. Without these, the work is meaningless.

Response: Thanks for your insightful suggestion. Compared to the previous reoprts to synthesize dibenzothiazepine derivatives in entry 1, our research used N, N’-dimethylethane-1,2-diamine (DMEDA) in small quantities as bifunctional-reagent, which worked as ligand and reductant, the reaction range is wide (the reaction substrates 2-iodobenzaldehydes/2-iodoacetophenones and 2-aminobenzenethiols/2,2'-disulfanediyldianilines could cross-react with each other) and the reaction conditions were simplified. We added the gram-scale (5 mmol scale) experiment to repeat the reaction in the manuscript, and the small amount of bifunctional-reagent DMEDA could be recycled in 71% by distillation of the reaction mixtrue.

  1. Scheme 1 includes a bromine derivative, but the entire work is based on iodine derivatives. In the case of bromine, there is only one example that has a low yield. In this case, the result does not make sense, because the cited literature presents similar fluorine, chlorine, and bromine derivatives with good yields.

Response: Thanks for your insightful suggestion. We deleted the example of bromine derivative in Scheme 1, and the corresponding revision was made in the manuscript.

  1. The decryption of DMEDA looks necessary in the main text of the article, at the first mention.

Response: Thanks for your insightful suggestion. We added some description of the importance of DMEDA in the Result Section.

  1. If the DMEDA is a bifunctional reagent, its role must be described. Moreover, this reagent is presented as a key in a catalytic process including the title. The acidic catalysis is more typical of the Schiff reaction. The entire process is not entirely clear and a plausible mechanism is suggested to be given. In addition, why are molecular sieves used? It is necessary to show the results of the reaction without cucl2 in the presence of molecular sieves.

Response: Thanks for your insightful suggestion. A possible mechanism and the corresponding description were added for the reaction in the manuscript, we found that the bifunctional-reagent DMEDA could be worked as ligand and reductant. The use of molecular sieves can absorb the H2O produced by the condensation reaction, so that the equilibrium of the reaction moves toward the product. The experiment was add, and the reaction yield was reduced Without the 4 Å molecular sieves (Table 1, entry 12).

  1. Authors should check references, for example, ref. 15 - journal name should be Chem. Eur. J. In addition, check the writing of the Refs. 30.31.

Response: We have revised relative the errors in the reference section of the manuscript.

Round 2

Reviewer 1 Report

I am still not convinced about the novelty of this work. Yes, it is true that N, N’-dimethylethane-1,2-diamine (DMEDA) acts as a bifunctional-reagent, which worked as ligand and reductant. If you compare with the reported study (Sharma et al. (Ref 11)) they used L-Proline as a ligand in catalytic amount (20 mol%). In Sharma et al., they performed the coupling with chloro derivatives whereas here only iodo derivatives are used. Does the authors have tried any chlorobenzaldehydes for coupling reaction?

In Table 2, scheme the temperature should be 110 deg not 10 deg.

Reviewer 3 Report

As was suggested previously, the authors should improve the introduction part. A similar text from the reply to the reviewer's comments should be given in the article. Scheme 1 should be modified because "entry 1" (previous works) and entry "This work" possess the same reactions. Authors should demonstrate in the main text as well as in the scheme the differences. Without these modifications, the novelty and scientific soundness of the work raises serious doubts